# Time-Kill Analysis of Canine Skin Pathogens: A Comparison of Pradofloxacin and Marbofloxacin

**DOI:** 10.3390/antibiotics12101548

**Published:** 2023-10-17

**Authors:** Stefano Azzariti, Andrew Mead, Pierre-Louis Toutain, Ross Bond, Ludovic Pelligand

**Affiliations:** 1Department of Comparative Biomedical Sciences, Royal Veterinary College, Hawkshead Lane, North Mymms, Hatfield AL9 7TA, UK; sazzariti@rvc.ac.uk (S.A.); amead@rvc.ac.uk (A.M.); pltoutain@rvc.ac.uk (P.-L.T.); 2INTHERES, Université de Toulouse, INRAE, Ecole Nationale Vétérinaire de Toulouse, 23 chemin des Capelles-BP 87614, CEDEX 03, 31076 Toulouse, France; 3Department of Clinical Sciences and Services, Royal Veterinary College, Hawkshead Lane, North Mymms, Hatfield AL9 7TA, UK; rbond@rvc.ac.uk

**Keywords:** canine pyoderma, *Staphylococcus* spp., *Escherichia coli*, fluoroquinolones, MIC, mathematical modelling, semi-mechanistic model, PK/PD index, dose fractionation, resistance

## Abstract

Time-kill curves (TKCs) are more informative compared with the use of minimum inhibitory concentration (MIC) as they allow the capture of bacterial growth and the development of drug killing rates over time, which allows to compute key pharmacodynamic (PD) parameters. Our study aimed, using a semi-mechanistic mathematical model, to estimate the best pharmacokinetic/pharmacodynamic (PK/PD) indices (ƒAUC/MIC or %ƒT > MIC) for the prediction of clinical efficacy of veterinary FQs in *Staphylococcus pseudintermedius*, *Staphylococcus aureus*, and *Escherichia coli* collected from canine pyoderma cases with a focus on the comparison between marbofloxacin and pradofloxacin. Eight TCKs for each bacterial species (4 susceptible and 4 resistant) were analysed in duplicate. The best PK/PD index was ƒAUC_24h_/MIC in both staphylococci and *E. coli*. For staphylococci, values of 25–40 h were necessary to achieve a bactericidal effect, whereas the calculated values (25–35 h) for *E. coli* were lower than those predicting a positive clinical outcome (100–120 h) in murine models. Pradofloxacin showed a higher potency (lower EC_50_) in comparison with marbofloxacin. However, no difference in terms of a maximal possible pharmacological killing rate (E_max_) was observed. Taking into account in vivo exposure at the recommended dosage regimen (3 and 2 mg/kg for pradofloxacin and marbofloxacin, respectively), the overall killing rates (K_drug_) computed were also similar in most instances.

## 1. Introduction

The use of minimum inhibitory concentrations (MICs) is pivotal in the evaluation of the pharmacodynamics (PD) of antimicrobials. Its measurement has been standardised, and the clinical breakpoints are the MIC values that allow the interpretation of antimicrobial susceptibility tests (ASTs) [1]. Nevertheless, MICs have limitations as these values represents a single terminal observation at about 24 h after initiation of the test, and these data do not inform on the time course of the bacterial count from an initial inoculum [2,3,4,5].

Time-kill curves (TKCs) or time-kill assays are more informative and quantify the chronological change in bacterial populations from a standardized initial inoculum exposed to one or multiple MICs. The antibiotic effect is measured in relation to growth and death of bacteria over time and is not only a snapshot of a single antimicrobial drug (AMD) concentration with its net effect [6,7]. For TKCs, a given bacterial inoculum can be exposed either to a static nominal antibiotic concentration or to concentrations that vary over time (dynamic time-kill curves, e.g., hollow fibre infection model) [8].

The PD-based informative value of TKCs can be revealed by mathematical modelling, which consists of linking bacterial exposure to an antibiotic concentration (constant or variable) to an observed effect through a pharmacokinetic/pharmacodynamic (PK/PD) model. The classical PD model is the E_max_ model, which allows the estimation of two genuine PD parameters separately: a maximum possible effect noted as E_max_ and the antibiotic concentration giving an effect equal to E_max_/2 noted as EC_50_ (effective 50% concentration). E_max_ is a measure of antibiotic pharmacological efficacy, and EC_50_ is a measure of antibiotic potency. A more advanced model is the Hill model; that is, the E_max_ model with a supplementary parameter (gamma) describing the slope of a sigmoidal concentration-effect curve, thus reflecting the steepness or sensitivity of the concentration-effect relationship [9].

Mathematical modelling of time-kill curve data also provides useful insight into the selection of the best PK/PD index to predict clinical efficacy, e.g., ƒAUC/MIC or %ƒT > MIC (with ƒ indicating the free concentration of the antimicrobial drug used for computation). These two PK/PD indices combine drug exposure and MIC values with the drug-related response over time and at specific time points [8]. The advantage of the in silico method also relies on the replacement of experimental animals, such rodents, for dose fractionation studies [10]. PD and PK data can be integrated from independent (or the same) studies, with the aim to administer a dose that achieves a plasma concentration able to reach or exceed an average free plasma concentration that equals the MIC (ƒAUC/MIC) or to achieve a free plasma concentration above the MIC for a given fraction of the dosing interval (%ƒT > MIC).

Several research groups have proposed frameworks for more advanced mathematical modelling of antimicrobial drugs on bacterial populations using semi-mechanistic PK/PD models. These combine biological and mechanistic knowledge, where observed data guide the model structure and parameter estimates. Models describe (1) bacterial growth and natural death, (2) drug effects, (3) regrowth and resistance emergence, and (4) antibiotic combinations and their interactions [8]. These models have been implemented in veterinary medicine [7,11,12]. With regard to veterinary fluoroquinolones (FQs) and canine skin pathogens, different studies have evaluated their killing effects in comparison with other antimicrobial classes using standard kill curves without modelling [13,14] or a simple inhibitory I_max_ model using only the final bacterial counts at 24 h [15]. Moreover, in vitro dynamic studies showed that at concentrations associated with standard oral doses of marbofloxacin (2 mg/kg) and pradofloxacin (3 mg/kg), a higher and more sustained bactericidal effect was observed with pradofloxacin [16].

Accordingly, the objective of our study was to (1) implement a semi-mechanistic mathematical model to time-kill analysis, (2) to estimate bacterial growth parameters and PD parameters, and (3) obtain the best PK/PD indices for the prediction of clinical efficacy of two veterinary FQs against three bacterial species, namely *S. aureus*, *S. pseudintermedius*, and *E. coli*, through dose-fractionation studies.

We hypothesised that (i) pradofloxacin was more potent (EC_50_ lower) and potentially more bactericidal (higher E_max_) in comparison to marbofloxacin and that (ii) predicted clinical efficacy at steady-state dosing would, however, be less evident between the two FQs due to differences in plasma exposure that favour marbofloxacin due to lower plasma clearance [17,18].

## 2. Results

### 2.1. Descriptive Time-Kill Analysis

In susceptible isolates, the highest bactericidal effect (expressed as 3 log_10_ reduction), was achieved at 8 times the MIC within 4–8 h in *S. pseudintermedius* (median 8 h for marbofloxacin and 6 h for pradofloxacin), 4–24 h in *S. aureus* (median 6 h for both drugs), and 0.5–4 h in *E. coli* (1.25 h for marbofloxacin and 1.5 h for pradofloxacin). In resistant isolates, the highest bactericidal effect was achieved at 8 times the MIC within 4–8 h for *S. pseudintermedius* (median 8 h for both drugs), 4–24 h for *S. aureus* (median of 24 h for marbofloxacin and 8 h for pradofloxacin), and 4–24 h for *E. coli* (median 7 h for marbofloxacin and 4 h for pradofloxacin).

### 2.2. Time-Kill Curve Mathematical Modelling

Semi-mechanistic models were used for the TKC analysis and modelling as proposed by Nielsen and Friberg [19] (Figure 1) for staphylococci, and Campion et al. [20] (Figure 2) for *E. coli*.

Parameters relating to growth of the bacterial system were shared for both drugs and are summarized in Table 1, with drug PD parameters provided in Table 2, Table 3 and Table 4. Visual predictive check (VPC; Appendix A) showed good predictability of the model with exposure of both drugs and their relative concentrations (as multiples of MICs) tested. Goodness of fit plots (PRED, IPRED, CWRES and IWRES) were appropriate (Appendix A).

Average drug concentrations and predicted in vivo drug effects are presented in Table 5. Critical values of PK/PD index are presented in Appendix A. For all dose fractionations, ƒAUC_PK_0–24h_/MIC outperformed ƒ%T > MIC as the best PK/PD index for both FQs against either of the three bacterial species.

### 2.3. Time-Kill Curve Analysis

*S. pseudintermedius*: The maximal growth rate (K_growthmax_) was at 1.41 h^−1^, which means that doubling the population size under the maximum growth condition requires 0.71 h or 42 min (Table 1). The natural death rate (K_death_), which naturally occurs in all bacterial populations at a constant rate and as a “net kill” during the stationary phase when bacteria are not growing, was fixed at 0.179 h^−1^, corresponding to an average lifespan of 335 min for a bacterium. Alpha, the rate constant reflecting the delay required to achieve a maximal steady-state growth rate, was estimated as 0.39 h^−1^, corresponding to an average delay of 154 min.

The median maximum bacterial density reached (B_max_) was 6.02 × 10^9^ CFU/mL.

The pradofloxacin-induced maximal bacterial killing rate (E_max_) was 2.23 h^−1^ (or equivalently a mean killing rate of 27 min) for susceptible and 1.80 h^−1^ (33 min) for resistant isolates (Table 2), which resulted in a 13.4-fold and 11.0-fold increase in the overall death rate, respectively. The marbofloxacin-induced maximal bacterial killing rate (E_max_) was 1.85 h^−1^ (32 min) for susceptible and 1.64 h^−1^ (36 min) for marbofloxacin-resistant isolates, which resulted in a 11.3-fold and 10.2-fold increase in the overall death rate.

For the susceptible isolates, estimates of potency EC_50_ (concentration necessary to achieve 50% of the maximal effect) ranged from 0.170 to 0.49 µg/mL for marbofloxacin and 0.031 to 0.041 µg/mL for pradofloxacin. For the resistant isolates, potency estimates ranged from 2.63 to 22.83 µg/mL for marbofloxacin and 0.25 to 1.49 µg/mL for pradofloxacin.

A high correlation was observed between the experimental and estimated MICs for both marbofloxacin (R^2^ = 0.9536) and pradofloxacin (R^2^ = 0.9925) (Appendix A).

*S. aureus*: K_growthmax_ was calculated as 1.36 h^−1^, which yielded an average maximum growth rate of 44 min (Table 1), versus a natural death rate (K_death_) fixed at 0.179 h^−1^ (335 min). Alpha was estimated as 0.77 h^−1^, which is equivalent to an average delay of 78 min. B_max_ (maximum inoculum reached) was 6.60 × 10^9^ CFU/mL.

E_max_ was similar in susceptible and resistant isolates for both drugs at 2.17 h^−1^ (28 min) for pradofloxacin and 1.97 h^−1^ (30 min) for marbofloxacin, which resulted in 13.1-fold and 11.9-fold increases in the overall death rate, respectively.

For the susceptible isolates, estimates of potency EC_50_ values ranged from 0.25 to 0.31 µg/mL for marbofloxacin and 0.031 to 0.072 µg/mL for pradofloxacin. For the resistant isolates, potency estimates ranged from 15.47 to 60.60 µg/mL for marbofloxacin and 1.29 to 4.46 µg/mL for pradofloxacin (Table 3). A high correlation was observed between measured and estimated MICs for both drugs (R^2^ = 0.9880 marbofloxacin, R^2^ = 0.9501 pradofloxacin) (Appendix A).

*E. coli*: A pre-existing heterogenous population model was adopted for *E. coli* and greatly improved model fitting compared to the basic model (Figure 2). It included two initial bacterial subpopulations noted S1 (dominant susceptible) and S2 (subdominant less susceptible) (Figure 2), which can be estimated both in FQ-susceptible and FQ-resistant isolates. This was either explained by spontaneous mutations in the absence of FQ or by an increase in the rate of mutation when exposed to concentrations of FQ below or around the MIC. Alpha, the rate constant that reflects a possible growth delay of *E. coli* in the test medium, was not included in the model as no delay was observed in TKCs in comparison with staphylococci; removing this parameter was not deleterious to the model fit.

K_growthmax_ was estimated as 2.00 h^−1^, which is equivalent to an average maximum growth rate of 30 min (Table 1), versus a natural death rate (K_death_) fixed at 0.179 h^−1^ (335 min). B_max_ was estimated as 6.84 × 10^9^ CFU/mL. The pradofloxacin-induced maximal bacterial killing rate (E_max_) was 8.73 h^−1^ (6.9 min) for susceptible and 3.11 h^−1^ (19 min) for resistant bacteria (Table 4), which resulted in 49-fold and 18.4-fold increases in the overall death rate, respectively. The marbofloxacin-induced maximal bacterial killing rate (E_max_) was 17.1 h^−1^ (3.5 min) for susceptible and 2.85 h^−1^ (21 min) for resistant isolates, which resulted in 96.0-fold and 16.9-fold increases in the overall death rate, respectively.

For the susceptible isolates, estimates of potency for S1 (EC_50_) ranged from 0.033 to 0.076 µg/mL for pradofloxacin and from 0.119 to 0.642 µg/mL for marbofloxacin. For the resistant bacteria, EC_50_ estimates for S1 ranged from 1.81 to 13.38 µg/mL for pradofloxacin and from 4.03 to 23.55 µg/mL for marbofloxacin. For the less susceptible S2 population, EC_50_ values were 1.97 and 1.67-fold higher that the EC_50_ values of S1 for pradofloxacin and marbofloxacin, respectively.

The proportion of S1 (named F or distribution factor) was estimated as 0.99. When S1 was equal to the initial inoculum of 5 × 10^5^ CFU/mL, S2 was negligible (>1 CFU/mL), but resistant mutants appeared when exposed to FQ concentrations lower than the MIC.

A good correlation was observed between the experimental and estimated MIC values in both FQs (R^2^ = 0.986 S1 and R^2^ = 0.9887 S2 for marbofloxacin, R^2^ = 0.9975 S1 and R^2^ = 0.9973 S2 for pradofloxacin) (Appendix A).

### 2.4. Comparison of Predicted In Vivo Effects

*S. pseudintermedius*: Taking into account in vivo exposure to the recommended dosage regimen, the average drug killing effect (K_drug_) at steady state in a selected susceptible isolate (MRSP 41) was 1.82 h^−1^ (33 min) for marbofloxacin and 2.21 h^−1^ (27 min) for pradofloxacin. The average drug effect at steady state in a selected resistant isolate (MRSP 67) was 0.08 h^−1^ (750 min) for marbofloxacin and 1.43 h^−1^ (42 min) for pradofloxacin (Table 5). No differences in terms of killing effect were observed for marbofloxacin between the two assessed protein bindings (9.1% [22] and 25% [23]).

*S. aureus*: Taking into account in vivo exposure to the recommended dosage regimen for a selected susceptible isolate (MSSA B98), the time-average drug effect (K_drug_) at steady state was 1.78 h^−1^ (34 min) for marbofloxacin and 2.15 h^−1^ (28 min) for pradofloxacin. The average drug effect at steady state in a selected resistant isolate (MSSA B53) was 0.0021 h^−1^ (i.e., negligible) for marbofloxacin and 0.28 h^−1^ (214 min) for pradofloxacin (Table 5). No differences were observed for marbofloxacin between the two assessed protein binding values (9.1% [22] and 25% [23]).

*E. coli*: Taking into account in vivo exposure to the recommended dosage regimen for a selected susceptible isolate (*E. coli* 14L-1510), the time-average drug effect (K_drug_) at steady state was 15.78 h^−1^ (3.8 min) for marbofloxacin and 8.23 h^−1^ (7.3 min) for pradofloxacin. The average drug effect at steady state in a selected resistant isolate (*E. coli* 12L-2253) was 0.0054 h^−1^ (i.e., negligible) for marbofloxacin and 0.11 h^−1^ (545 min) for pradofloxacin (Table 5). No differences were observed for marbofloxacin between the two assessed protein binding values used (9.1% [22] and 25% [23]).

### 2.5. Confirmation of PK/PD Index Target Value

*S. pseudintermedius*: For a typical susceptible bacterium, such as MRSP 41 (MIC marbofloxacin 0.18 µg/mL and pradofloxacin 0.025 µg/mL), target ƒAUC_PK_0–24h_/MIC values to achieve 90% E_max_ values were 35.9 and 35.7 h, corresponding to average free plasma concentrations of 0.26 and 0.037 µg/mL for marbofloxacin and pradofloxacin, respectively (Appendix A). No differences were observed for marbofloxacin between the two assessed protein binding values (9.1% [22] and 25% [23]).

*S. aureus*: For a typical susceptible bacterium, such as MSSA B98 (MIC marbofloxacin 0.35 µg/mL and pradofloxacin 0.05 µg/mL), target ƒAUC_PK_0–24h_/MIC values to achieve 90% E_max_ values were 31.7 and 44.9 h, corresponding to average free plasma concentrations of 0.46 and 0.094 µg/mL for marbofloxacin and pradofloxacin, respectively (Appendix A). No differences were observed for marbofloxacin between the two assessed protein binding used (9.1% [22] and 25% [23]).

*E. coli*: For a typical susceptible bacterium, such as *E. coli* 14L-1510 (MIC marbofloxacin 0.35 µg/mL and pradofloxacin 0.05 µg/mL), target ƒAUC_PK_0–24h_/MIC values to achieve 90% E_max_ values were 26.9 and 26.3 h, corresponding to average free plasma concentrations of 0.028 and 0.024 µg/mL for marbofloxacin and pradofloxacin, respectively (Appendix A). No differences were observed for marbofloxacin between the two assessed protein bindings used (9.1% [22] and 25% [23]).

## 3. Discussion

This is the first study that compared two veterinary FQs used in veterinary dermatology using a PK/PD mathematical modelling approach. Our results showed that, at their licensed doses, the two investigated FQs do not show substantial differences in comparison with the predicted clinical efficacy against *S. pseudintermedius*, *S. aureus*, and susceptible *E. coli* collected from canine pyoderma. Moreover, as expected from the MIC measurements, TKC modelling showed that pradofloxacin is more potent than marbofloxacin (i.e., had lower EC_50_). However, their pharmacological efficacies measured from their maximum killing rates were similar. These two PD parameters generated from TKC analysis are not capable on their own of ranking these two antibiotics in terms of expected clinical efficiency. Thus, further integration into a PK/PD type index is necessary to obtain an overall assessment, which will take into account the administered doses and the internal exposure of the two drugs for the registered dosage regimen.

The main objective of the study was to compare two FQs used in veterinary dermatology, namely marbofloxacin and pradofloxacin. However, the findings from the study do not allow conclusions to be drawn on the use of the antibiotics in veterinary practice, for which randomised clinical trials represent the gold standard when comparing two antibiotics and their clinical outcome. These are not common in veterinary medicine, and, to our knowledge, clinical trials have not been conducted using these two antibiotics in the treatment of canine skin infections. It is therefore necessary to consider other surrogate criteria and to speculate on comparisons between the two antibiotics in terms of clinical efficacy. MIC is often implemented in these cases for the target pathogens. With regards to the two analysed FQs, pradofloxacin shows a much lower value in comparison with marbofloxacin, which could be erroneously interpreted as a clinical advantage. Indeed, MIC is only a hybrid variable that reflects standardised in vitro conditions of drug potency and efficacy and conclusions on the superiority of antibiotics in terms of clinical efficacy cannot be drawn from MIC comparison.

In this regards, it should be noted that the potency of a drug and its pharmacological efficacy are two pharmacological properties and that the efficacy measured under controlled in vitro conditions is only one of the elements that determines the overall clinical efficacy. Other factors determining clinical efficiency include the PK properties of the antibiotic and the selected dosage regimen.

Marbofloxacin and pradofloxacin were compared using a dynamic model that reproduced the in vivo plasma concentration profile obtained from licensed oral doses in dogs (2 and 3 mg/kg for marbofloxacin and pradofloxacin, respectively) in the test medium [16]. The author concluded that pradofloxacin possesses a greater bactericidal effect against both *S. pseudintermedius* and *S. aureus* in comparison with marbofloxacin. The results were attributable to higher ƒAUC_24h_/MIC values resulting from pradofloxacin that had a more rapid decrease in the initial inoculum of 5 × 10^7^ CFU/mL. However, as mentioned in the Introduction section, this type of experimental protocol does not characterise the genuine pharmacodynamic properties of an antibiotic, and the results obtained are limited to single plasma concentration profiles, which have been reproduced for a given tested pathogen. Hence, our study aimed to quantify and characterize the main PD properties of marbofloxacin and pradofloxacin in order to predict their comparative efficiency. TKC assays are the most informative in vitro approach to estimate the PD properties of an antibiotic provided that the results obtained are modelled with a model in which potency and efficacy are explicitly estimable from the temporal dynamics of bactericidal activity. This is the case for the various semi-mechanistic models recently reviewed by Minichmayr et al. [8], and this is the first study that compared the PK/PD of marbofloxacin and pradofloxacin using TKC modelling.

ƒAUC_24h_/MIC has been widely adopted in the literature for all FQs, including marbofloxacin and pradofloxacin [24,25]. However, the values needed to achieve bacterial eradication and clinical efficacy have never been investigated in veterinary medicine. In humans, an ƒAUC_24h_/MIC of 125 h is often reported following clinical data [25]. For instance, an average free plasma concentration of a selected FQ over a 24-h dosing interval equal to 5 times the MIC should be recommended for therapeutic success. This leads to the selection of rather high doses of FQs, and alternative and more conservative lower index values have been proposed, in line with prudent use of veterinary antimicrobials as suggested by WHO guidelines [26]. Thus, achieving an ƒAUC_24h_/MIC at 48 h makes it possible to reduce the dose by a factor of 2.6, in comparison with an ƒAUC_24h_/MIC of 125 h. This can be achieved simply using the model that was developed to estimate PD parameters by replacing the static KC concentrations by the time course of free plasma concentrations expected for different dosing regimens. This approach allowed us to confirm that *f*AUC/MIC was the PK/PD index most predictive of the bactericidal effect for both drugs. In staphylococci, an ƒAUC_24h_/MIC of approximately 25–45 h was necessary to achieve 90% of the in silico inhibitory effect in both susceptible and resistant isolates, which is in line with previously reported data from both in vivo and in vitro studies [24,25,27,28]. However, Lorenzutti et al. [12] showed that in *S. aureus*, a PK/PD index of at least 120 h is necessary to obtain 50% of the inhibitory effect when high inocula are used. This value, which is similar to the 125 h often recommended by default in human medicine, can be explained by the fact that high inocula already contain first-step mutant bacteria with a higher MIC. Thus, the likelihood of selecting subpopulations of first-step mutants is reduced by aiming for concentrations clearly higher than the MIC of the initial wild population. This is in line with results obtained in the murine thigh infection model of Ferran et al. [29], which demonstrated that the likelihood of resistance emergence to marbofloxacin is influenced by the inoculum size and pre-existing mutants before any antimicrobial treatments are administered. Likely for the same reason, the ƒAUC_24h_/MIC values obtained from *E. coli* in our study are lower than those necessary to achieve a bacteriological cure at approximately 100–125 h and are obtained from human clinical trials or animal infection models. Indeed, ex vivo studies on veterinary FQs support that ƒAUC_24h_/MIC values lower than 100 h can achieve bactericidal activity (3log10 reduction) or total eradication in Gram-negative bacteria from pigs [30] or calves [31,32,33]. However, dynamic kill studies, which can mimic the drug effects in patients with immunosuppression or critical illnesses, have demonstrated with both human [34] and veterinary [35] FQs that ƒAUC_24h_/MIC values lower than 100 h were associated with regrowth and/or resistance emergence.

Moreover, a review by Wright et al. [25] showed that ƒAUC_24h_/MIC values of 30–35 h are necessary to achieve a clinical cure. It is relevant for veterinary medicine to note that for *S. pseudintermedius* and *S. aureus* did not show any reduced susceptibility within 24 h both in the presence and absence of FQs (pradofloxacin and marbofloxacin) [16].

The last element to be taken into consideration when comparing the two FQs relates to their PK properties and their dosage regimen as recommended in their marketing authorizations. It would have been advantageous to have had access to PK raw data (plasma concentrations) for both marbofloxacin and pradofloxacin of different origins and to reanalyse them with a non-linear mixed Effect Model (NLMEM) as previously shown with different class of antibiotics [7], despite the commercial sensitivities.

Considering the significant differences in clearance and plasma protein binding between the two substances and taking into account the recommended dosing regimens, it appears that the internal exposure to free concentrations of marbofloxacin under steady-state conditions was 2.4-fold higher than that of pradofloxacin with equilibrium free concentrations of 0.76 and 0.32 mg/L for marbofloxacin and pradofloxacin, respectively.

Future experiments should involve the acquisition of that information to propose a clinical breakpoint using Monte Carlo simulation. For instance, Yohannes et al. [15] used a PK/PD model to investigate whether intramuscular and intravenous doses of 2 mg/kg marbofloxacin were able to achieve adequate PK/PD indices necessary to obtain bacterial eradication in *S. pseudintermedius*, both in vitro and ex vivo (serum). PK/PD modelling resulting from real plasma concentrations, coupled with time-kill analysis, showed that the doses could not achieve the target AUC/MIC necessary to eradicate *S. pseudintermedius*; therefore, different dose regimens should be explored.

## 4. Materials and Methods

### 4.1. Selection of Bacterial Isolates and Antimicrobial Susceptibility

Twenty-four isolates obtained from cases of canine pyoderma were included in the study, comprising 8 *S. pseudintermedius*, 8 *S. aureus* and 8 *E. coli* isolates. We selected 1 pool of 4 marbofloxacin-susceptible isolates and 1 pool of 4 marbofloxacin-resistant isolates (Appendix A). Resistant isolates were screened for chromosomal DNA gyrase (*gyrA*) and topoisomerase IV (*grlA*/*parC*) mutations by PCR as previously described [36,37,38]. *E. coli* was also screened for plasmid-mediated quinolone resistance (PMQR) genes [39,40]. Susceptible isolates were considered as wild-type bacteria (WT) (Appendix A). Equal proportions of methicillin susceptible (4 MSSA, 4 MSSP) and resistant (4 MRSA, 4 MRSP) staphylococcal isolates were selected.

### 4.2. MIC Measurement and Time-Kill Curve Technique

For the TKC study, MICs of marbofloxacin and pradofloxacin were initially obtained using the broth microdilution method according to CLSI guidelines [41]. Subsequently, a more accurate MIC measurement called “five series of overlapping dilutions” was adopted in this study. The method was developed by Aliabadi and Lees [42], and compared with the standard MIC measurement, it has the advantage of reducing the inaccuracy of the 2-fold MIC measurement from 100% to a maximum of 20%.

Isolates were thawed from storage in brain heart infusion (BHI, Oxoid, Basingstoke, UK) + 25% glycerol at −80 °C and subcultured three times on blood agar to promote optimal growth according to CLSI guidelines [43]. MR isolates were subcultured two times on Mannitol salt agar (Oxoid, Basingstoke, UK) + 6 mg/L oxacillin (MS+) and subsequently on BA to ensure that methicillin resistance (MR) was preserved. Bacterial density was adjusted to equivalence with 0.5 McFarland standard (approx. 1–2 × 10^8^ CFU/mL) using DensiCheck^®®^ (Biomérieux, Marcy L’Étoile, France) and diluted 100-fold to achieve a 10^6^ CFU/mL final suspension. Then, 50 µL of bacterial suspension was added to each well of a 96-well plate containing 50 µL of prewarmed FQ-containing cation-adjusted Mueller-Hinton broth (CAMHB, Merck, Steinheim, Germany) to yield a final inoculum of 5 × 10^5^ CFU/mL exposed to 0×, 0.5×, 1×, 2×, 4× and 8× of the MIC. Plates were incubated at 37 °C for 24 h. At 0, 0.5, 1, 2, 4, 8 and 24 h of incubation, 25 µL from each of the wells was serially 10-fold diluted with phosphate-buffered saline (PBS, Fisher, Geel, Belgium) up to 10^−8^. After dilution, 25 µL spots were applied onto square petri dishes containing Mueller-Hinton agar (MHA, Merck, Steinheim, Germany), left to dry in a microbiology cabinet and then incubated at for 16–24 h at 37 °C. After incubation, cell counts expressed in CFU/mL were back calculated from the lowest dilution with approximately 3–30 CFU per spot. The limit of enumeration, defined as the lowest number of countable cells, was set at 40 CFU/mL (one or no colonies in a 25 µL spot at the lowest dilution). Each experiment was conducted in duplicate on separate days.

Results were log_10_ transformed and plotted with time (h) on the x axis and bacterial count (CFU/mL) on the y axis.

### 4.3. Pharmacodynamic Data Analysis and Modelling

Data from TKCs were analysed using the Phoenix 8.3.0.5005 software package (Certara, Princeton, NJ, USA). A semi-mechanistic model proposed by Nielsen and Frieberg [19] was initially adopted for this study. The model included two compartments: the “S” containing susceptible bacteria, and a second compartment “P”, containing persisters, which are non-growing and drug-insensitive bacteria (Figure 1).

Mathematical modelling, including a bacterial growth model, drug effect and secondary parameters (MICs), were obtained following a method previously described by Pelligand et al. [44] (Appendix A).

### 4.4. Pre-Existing Heterogenous Population Model

Since the initial model proposed by Nielsen and Friberg [19] did not capture the regrowth of *E. coli* exposed to 1 × MIC, we hypothesised that pre-existing first-step mutants were present within the total population of *E. coli*., as described by Campion et al. [20]. The ratio between first-step mutants and the total bacterial population has been shown to be approximately 10^−8^ to 10^−9^ based on growth curves [34]. Thus, we concluded that the initial population (starting inoculum) consisted of two subpopulations, representing a heterogenous bacterial population with a proportion (F1) of bacteria being a highly susceptible dominant population (S1) and the remaining sub-dominant population (S2) having a lower susceptibility. F1 was estimated by the model illustrated in Figure 2 and as shown by Mead et al. [11].

### 4.5. Covariate Analyses

To allow for parameter adjustment in relation to the initial FQ susceptibility of isolates (resistant versus non-resistant), a binary categorial covariate was tested during PD parameters estimation for system parameters common to both drugs (e.g., K_growthmax_, alpha, and B_max_) or drug-specific parameters for each drug (E_max_, EC_50_, and gamma). Selection of significant covariate effects was carried out using the Stepwise Phoenix tool. It involved stepwise forward and backward parallelized addition and deletion of covariate effects. Covariates were added progressively to determine if the goodness of fit improved compared to a predefined threshold in objective function value (OFV). In this case, the Bayesian information criterion (BIC) with thresholds of 6.635 and 10.828 points was used to add and remove a covariate, respectively. The covariate analysis was carried out to elucidate the origin of variability of resistant or susceptible isolates; hence, resistance vs susceptible was considered as a factor rather than a covariate.

### 4.6. Comparison of In Vivo Drug Effects Predicted from Licensed Dosage Regimens

Plasma concentration time curves were obtained using the Unscan it software (version 7.0, Silk Scientific, Provo, UT, USA) from the previously published PK studies where dogs received 2 mg/kg marbofloxacin [18] or 3 mg/kg pradofloxacin [21] (Appendix A). Non-compartmental analysis was carried out on free plasma concentrations in Phoenix^®®^ NLME^®®^ Protein, and binding of 36% [45] was considered for pradofloxacin. For marbofloxacin two different protein bindings (9.1% from product monograph [22] and 25% Bregante et al. [23] were explored. The PK parameters are presented in Appendix A (C_max_, T_max_, half-life, clearance and AUC_0–24h_ and AUC_0-infinity_). The average plasma concentrations after the first dose and at steady state were calculated by dividing the corresponding AUC (AUC_0–24h_ and AUC_0-infinity_, respectively) by 24 h [46].

Time-averaged drug effects (K_drug_, h^−1^) were computed from the Hill’s equation (Text S1, Equation (2)) and these average daily free concentrations to compare predicted in vivo effects (K_drug_) between FQs for each bacterial species (susceptible and resistant).

### 4.7. In Silico Dose Fractionation Experiments

We performed an in silico dose fractionation to confirm the nature of the best PK/PD index (ƒAUC_PK_0–24h_/MIC or %ƒT > MIC) by simulating the bacteriological response to a wide range of escalating dosing regimens in order to achieve no effect or maximum effects. In total, 12 escalating doses (including control curve which corresponds to no antibiotics administered) were included and administered:-As a single dose over 24 h-Split in 2 half-doses, given every 12 h-Split in 4 quarter-doses, given every 6 h

Similar to the predicted drug effect, two different protein bindings of marbofloxacin were taken into account.

This yielded a total of 34 dosage regimens for which we computed the area under the free FQ concentration curve over the MIC (ƒAUC_PK_0–24h_/MIC) and the percentage of time the free concentration exceeded the MIC (%ƒT > MIC). Effects on bacterial inoculum were evaluated up to 24 h, and the area under the bacterial concentration curve (AUC_bact-24h_) was computed. The bacterial count was set to 0 (eradication) as soon as the bacterial concentration reached the LOQ of the enumeration method (40 CFU/mL).

An inhibitory effect sigmoid PD model (I_max_) was implemented to fit the PK/PD index (%ƒT > MIC and ƒAUC__0–24h_/MIC) versus AUC_bact-24h_. The I_max_ model describes the inhibitory effect of a drug according to the following formula (Equation (1)):(1)$$\mathrm{Effect}=E_{0}-\frac{\left(I_{\max}\times {\mathrm{index}}^{\mathrm{slope}}\right)}{\left({\mathrm{index}}^{\mathrm{slope}}+{\mathrm{index}}_{50}^{\mathrm{slope}}\right)}$$

Here, effect (the observed effect), E_0_, and I_max_ are expressed in terms of Log_10_AUCbact_0–24h_. Specifically, E_0_ represents the maximal effect obtained for the control curve, I_max_ the amplitude of the maximal inhibitory effect, and E_0_-I_max_ represents the maximal possible inhibitory observed effect. Index_50_ is the magnitude of the PK/PD index (%ƒT > MIC or ƒAUC__0–24h_/MIC) that achieves 50% of I_max_, and slope is the steepness of the sigmoid curve. The best PK/PD index that predicted the bactericidal effect was assessed by regression analysis (R^2^), Akaike index criterion (AIC), and visual inspection of graphs (Figure 3). Moreover, Index_90%_ was calculated as the breakpoint value for the PK/PD index, which allowed calculation of the average free plasma concentration required over 24 h to achieve 90% of the maximal efficacy.

## 5. Conclusions

When recommended doses, clearance, and protein binding differences are considered, we estimated that pradofloxacin may have a higher drug effect than marbofloxacin in some situations. However, the limitation of TCKs relies on static concentrations over time, which cannot capture the time course of drug concentrations during antimicrobial treatment.

For all the three bacterial species investigated in the study, the predicted clinical outcome relies on ƒAUC/MIC, which is dependent on drug exposure to achieve bacterial eradication. However, when comparing the predicted clinical efficacy between the two FQs, no substantial differences were observed in the susceptible isolates that are of concern in clinical settings.

The advantage of pradofloxacin depends on its narrow mutant selection window (MSW), which should limit the time during which target drug concentrations are likely to select for first-step mutants. To explore this hypothesis, further investigations on the resistance emergence and optimal dosing regimens should be implemented with dynamic kill studies (e.g., hollow fibre infection model) and randomised blinded clinical trials to predict and achieve complete bacteriological eradication, together with the reduction of resistance emergence.

## Figures and Tables

**Figure 1 antibiotics-12-01548-f001:**
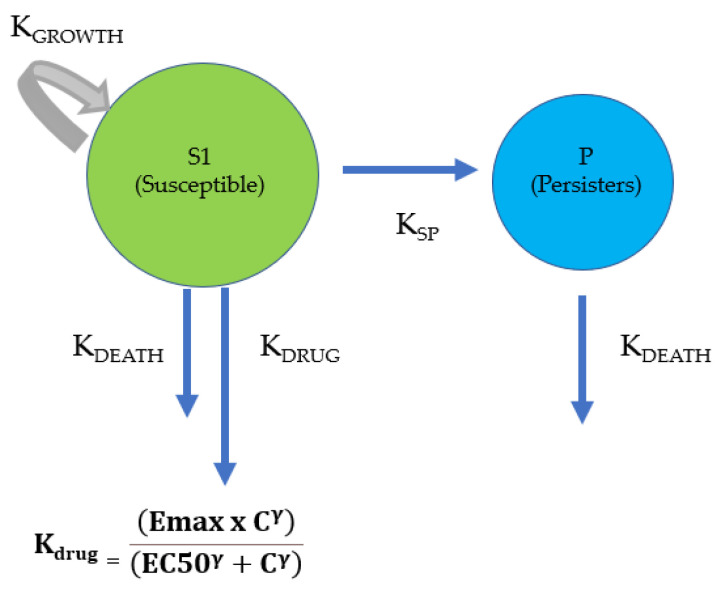
Semi-mechanistic model for TKC analysis and modelling proposed by Nielsen and Friberg [19]. The model comprises two compartments. The first compartment is “S”, which includes growing and drug-susceptible bacteria. Bacterial growth occurs at a constant rate K_growth_. Bacterial death (K_death_) naturally occurs during the stationary phase due to exhaustion of nutrients, or it is accelerated by the parallel drug effect (K_drug_). E_max_, EC_50_, and gamma are the PD parameters that describe efficacy (maximum effect achieved), potency (50% of E_max_), and sensitivity (steepness of the concentration-effect curve). The second compartment contains persisters (P), which are non-growing and non-drug-susceptible bacteria but eliminated according to K_death_. K_SP_ represents the irreversible constant rate between S and P compartments. K_death_ for persisters is the same as that noted for susceptible bacteria.

**Figure 2 antibiotics-12-01548-f002:**
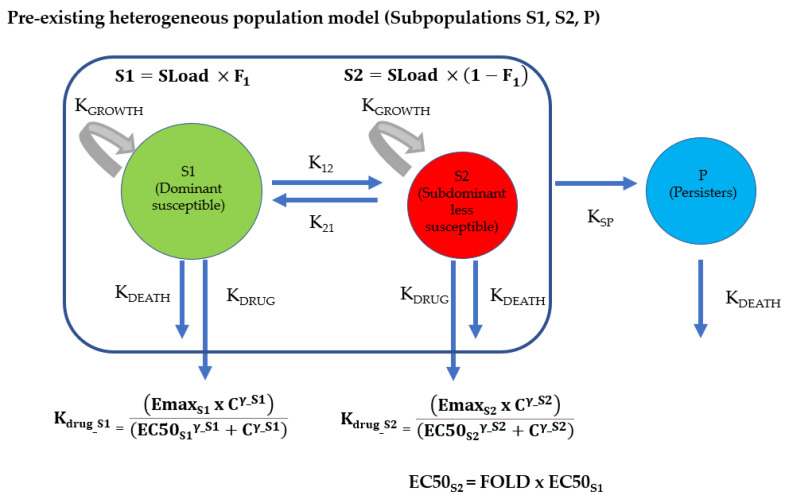
Schematic representation of pre-existing heterogenous population model adopted for *E. coli* time-kill analysis. The model comprises 3 subpopulations: S1, S2 and P. S1 is the dominant FQ-susceptible subpopulation, whereas S2 the subdominant less FQ-susceptible subpopulation This was either explained by spontaneous mutations in the absence of FQ or by an increase in the rate of the mutation when exposed to concentrations of FQ below or around the MIC. The S1 and S2 subpopulations have the same growth rate constant (K_growth_) and death rate (K_death_), but the drug killing effect is influenced by their different susceptibilities (K_drug_S1_ and K_drug_S2_). The system was considered to be already in equilibrium during the initial exposure to FQ, and the K_12_/K_21_ ratio was estimated by directly evaluating a distribution factor (F) of the population between the population S1 (F) and subpopulation S2 (1 − F) with F = (1 − F) × (K_12_/K_21_). The P compartment is represented by persisters, which do not grow but a have constant death rate (K_death_). K_SP_ is the irreversible transfer constant between S_1_ or S_2_ and P compartments. The potency ratio between S2 and S1 was estimated as FOLD = EC_50S2_/EC_50S1_ for each drug.

**Figure 3 antibiotics-12-01548-f003:**
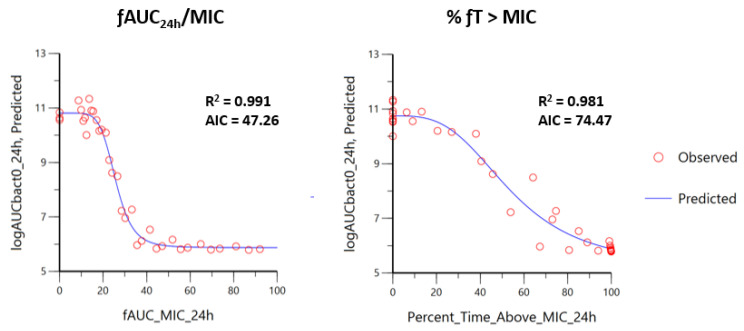
Comparison of fitting for prediction of log_10_AUCbact_0–24h_ (y axis) obtained from the Imax model, and ƒAUC_24h_/MIC ((**left plot**), x axis) or %ƒT > MIC ((**right plot**), x axis) in a selected isolate (MRSP 41) with an marbofloxacin MIC of 0.25 µg/mL. R^2^ is the coefficient of determination providing the percentage of variance explained by the model. AIC is the Akaike information criterion, which represents a measure of goodness of fit. In this case, ƒAUC_24h_/MIC represents the best PK/PD index with the highest R^2^ and the lowest AIC.

**Table 1 antibiotics-12-01548-t001:** Bacterial growth parameter estimates and confidence intervals in *S. pseudintermedius*, *S. aureus*, and *E. coli* collected from canine skin infection cases.

**Bacterial Growth System Parameters ** ** *S. pseudintermedius* **		**Bootstrap (*n* = 30)**		
	**Estimate**	**Median**	**2.5% CI**	**97.5% CI**
K_GROWTHMAX_ (h^−1^)	1.41	1.36	1.22	1.56
K_DEATH_ (h^−1^)	0.179	Fixed		
Alpha (h^−1^)	0.39	0.41	0.33	0.52
B_MAX_ (CFU/mL)	6.02 × 10^+09^	6.72 × 10^+09^	5.45 × 10^+09^	8.91 × 10^+09^
CV% B_MAX_ (inter-strain variability)	130%			
**Bacterial growth system parameters** ** *S. aureus* **		**Bootstrap (*n* = 30)**		
	**Estimate**	**Median**	**2.5% CI**	**97.5% CI**
K_GROWTHMAX_ (h^−1^)	1.36	1.36	1.23	1.50
K_DEATH_ (h^−1^)	0.179	Fixed		
Alpha (h^−1^)	0.77	0.79	0.58	9.99
B_MAX_ (CFU/mL)	6.60 × 10^+09^	7.16 × 10^+09^	5.59 × 10^+09^	1.07 × 10^+10^
CV% B_MAX_ (inter-strain variability)	124%			
**Bacterial growth System parameters** ** *E. coli* **		**Jacobian estimate**		
	**Estimate**	**CV%**	**2.5% CI**	**97.5% CI**
K_GROWTHMAX_ (h^−1^)	2.00	2.57	1.90	2.10
K_DEATH_ (h^−1^)	0.179	Fixed		
B_MAX_ (CFU/mL)	6.84 × 10^+09^	15.96	4.70 × 10^+09^	8.98 × 10^+09^

K_GROWTHMAX_: maximal growth rate; K_death_, natural death rate; Alpha, rate constant reflecting the delay required to achieve maximal steady-state growth rate; B_MAX_, maximum possible bacterial density.

**Table 2 antibiotics-12-01548-t002:** Pharmacodynamic parameters in 4 FQ-susceptible and 4 FQ-resistant strains of *S. pseudintermedius* collected from canine pyoderma or skin wound cases.

Isolate	PRADOFLOXACIN					MARBOFLOXACIN				
	Estimates		Bootstrap (*n* = 30)			Estimates		Bootstrap (*n* = 30)		
	Susceptible	Resistant	Median	2.5% CI	97.5% CI	Susceptible	Resistant	Median	2.5% CI	97.5% CI
	EC_50__Pradofloxacin (mg/L)					EC_50__Marbofloxacin (mg/L)				
MSSP_22219	0.037	-	0.036	0.029	0.049	0.17	-	0.18	0.15	0.21
MSSP_108	0.041	-	0.041	0.032	0.054	0.49	-	0.45	0.41	0.53
MRSP_1726	0.033	-	0.039	0.030	0.054	0.18	-	0.19	0.15	0.21
MRSP_41	0.031	-	0.034	0.026	0.048	0.18	-	0.17	0.12	0.19
MSSP_98	-	0.25	0.25	0.22	0.28	-	2.63	2.79	2.58	3.06
MSSP_115	-	0.82	0.82	0.69	0.90	-	8.41	8.58	7.70	11.38
MRSP_38	-	1.49	1.72	1.46	1.84	-	19.94	22.21	19.71	32.26
MRSP_67	-	1.21	1.30	1.07	1.64	-	22.83	24.00	22.02	25.42
	E_max__Pradofloxacin (h^−1^)					E_max__Marbofloxacin (h^−1^)				
	2.23	-	2.36	1.74	2.87	1.85	-	1.91	1.53	2.04
	-	1.80	1.72	1.54	2.03	-	1.64	1.61	1.44	1.86
	Gamma_Pradofloxacin (scalar)						Gamma_Marbofloxacin (scalar)			
	1.90		1.87	1.67	2.54	2.58		2.58	2.30	3.01

E_max_, maximal increase in killing rate in addition to K_DEATH_ (e.g., 2.23 h^−1^ = mean constant of 27 min); EC_50_, concentration required to achieve 50% of E_max_; Gamma is the Hill’s coefficient. Common E_max_ and gamma values were estimated for the four susceptible and the four resistant isolates.

**Table 3 antibiotics-12-01548-t003:** Pharmacodynamic parameters in 4 FQ-susceptible and 4 FQ-resistant strains of *S. aureus* collected from canine pyoderma or skin wound cases.

Isolate	PRADOFLOXACIN					MARBOFLOXACIN				
	Estimates		Bootstrap (*n* = 30)			Estimates		Bootstrap (*n* = 30)		
	Susceptible	Resistant	Median	2.5% CI	97.5% CI	Susceptible	Resistant	Median	2.5% CI	97.5% CI
	EC_50__Pradofloxacin (mg/L)					EC_50__Marbofloxacin (mg/L)				
MSSA_476	0.061	-	0.062	0.058	0.085	0.32	-	0.317	0.297	0.359
MSSA_B98	0.072	-	0.076	0.069	0.103	0.31	-	0.31	0.29	0.36
MRSA_A53	0.051	-	0.052	0.040	0.066	0.28	-	0.28	0.26	0.32
MRSA_A54	0.031	-	0.031	0.024	0.052	0.25	-	0.25	0.22	0.28
MSSA_B53	-	1.29	1.28	1.24	1.48	-	15.47	15.74	14.78	16.85
MSSA_B94	-	1.34	1.34	1.29	1.90	-	17.05	17.77	16.15	19.16
MRSA_A009	-	1.34	1.33	1.26	1.87	-	16.30	16.72	15.53	17.65
MRSA_A69	-	4.46	4.20	4.10	5.30	-	60.60	62.30	57.22	65.31
	E_max__Pradofloxacin (h^−1^)					E_max__Marbofloxacin (h^−1^)				
	2.17		2.30	1.88	2.69	1.97		2.01	1.67	2.23
	Gamma_Pradofloxacin (scalar)					Gamma_Marbofloxacin (scalar)				
	2.06		1.98	1.14	2.68	2.34		2.28	1.80	2.86

E_max_, maximal increase in killing rate in addition to K_DEATH_ (e.g., 2.17 h^−1^ = mean constant of 28 min); EC_50_, concentration required to achieve 50% of E_max_; Gamma is the Hill’s coefficient. Common E_max_ and gamma values were estimated for the four susceptible and the four resistant isolates.

**Table 4 antibiotics-12-01548-t004:** Pharmacodynamic parameters in 4 FQ-susceptible and 4 FQ-resistant *E. coli* collected from canine pyoderma or skin wound cases.

Isolate	PRADOFLOXACIN					MARBOFLOXACIN				
	Estimates		Precision of Estimates			Estimates		Precision of Estimates		
	Susceptible	Resistant	CV%	2.5% CI	97.5% CI	Susceptible	Resistant	CV%	2.5% CI	97.5% CI
	EC_50__Pradofloxacin_S1 (mg/L)					EC_50__Marbofloxacin_S1 (mg/L)				
*E. coli* 14L_1510	0.047	-	8.72	0.04	0.05	0.119	-	9.01	0.10	0.14
*E. coli* 16L_1242	0.052	-	9.13	0.04	0.06	0.178	-	9.80	0.14	0.21
*E. coli* 17L_0826	0.033	-	9.43	0.03	0.04	0.157	-	9.74	0.13	0.19
*E. coli* 17L_1562	0.076	-	9.29	0.06	0.09	0.642	-	9.88	0.52	0.77
*E. coli* 2443	-	1.81	5.62	1.61	2.01	-	4.03	4.40	3.69	4.38
*E. coli* 10L_2253	-	2.07	5.88	1.83	2.30	-	7.81	3.50	7.27	8.34
*E. coli* 10L_3690	-	7.43	6.69	6.45	8.41	-	23.55	4.55	21.45	25.66
*E. coli* 15L_3275	-	13.38	5.37	11.97	14.80	-	15.91	4.43	14.53	17.29
	EC_50__Pradofloxacin_S2 = 1.97 × EC_50__Pradofloxacin_S1 (mg/L)					EC_50__Marbofloxacin_S2 = 1.67 × EC_50__Marbofloxacin _S1 (mg/L)				
	E_max__ Pradofloxacin (h^−1^)					E_max__Marbofloxacin (h^−1^)				
	8.73	-	3.63	8.11	9.35	17.14	-	4.22	15.72	18.56
	-	3.11	3.03	2.93	3.30	-	2.85	2.64	2.71	3.00
	Gamma_Pradofloxacin (scalar)					Gamma_Marbofloxacin (scalar)				
	1.17		5.58	1.04	1.30	1.12		3.91	1.03	1.20
		2.37	10.08	1.90	2.84		2.80	8.22	2.35	3.25

For both susceptible and resistant strains, the initial inoculum is homogeneous with a population noted S1 present exclusively in the inoculum (F = 99.999998%). Upon exposure to the FQ, a resistant population (for the 4 isolates initially susceptible) or more resistant (for the 4 isolates initially already resistant) noted S2 will emerge during the test. S2 is at the origin of the regrowth phenomenon when the tested concentrations of FQ are between the MICs of S1 and those of S2, and this dynamic is captured by the model making it possible to estimate the EC_50_ of S1 and S2, the EC_50_ of S2 were estimated by a multiplying factor of S1. E_max_, maximal increase in the killing rate in addition to K_DEATH_ (e.g., 8.73 h^−1^ = mean constant of 6.9 min); EC_50__S1, concentration required to achieve 50% of E_max_ for the dominant more susceptible population S1; Gamma is the Hill’s coefficient; coefficient of variation (CV%) and confidence interval (CI) are represented in the Jacobian estimate and obtained using a simple run model. FOLD is a parameter that represents the potency ratio between the EC_50S2_ and EC_50S1_.

**Table 5 antibiotics-12-01548-t005:** Average free plasma concentrations (µg/mL) and predicted in vivo killing drug effect (K_drug_) after the first dose administration and at steady state. These parameters were measured after simulation of PK data obtained from free-plasma concentrations of 2 mg/kg marbofloxacin from Schneider, Thomas, Boisrame and Deleforge [18] and 3 mg/kg pradofloxacin from Hauschild, et al. [21]. With regards to marbofloxacin, 9.1% [22] and 25% (value in bold, [23]) protein binding were considered, and the corresponding average plasma concentration and K_drug_ values at the first dose and at steady state were calculated. Two representative isolates (susceptible and resistant) were chosen for each bacterial species (*S. pseudintermedius*, *S. aureus,* and *E. coli*).

Isolate	PRADOFLOXACIN					MARBOFLOXACIN				
		Average Free Plasma Concentration (µg/mL)		K_drug_ (h^−1^)			Average Free Plasma Concentration (µg/mL)		K_drug_ (h^−1^)	
	Experimental MIC (µg/mL)	First Dose	Steady State	First Dose	Steady State	Experimental MIC (µg/mL)	First Dose	Steady State	First Dose	Steady State
MSRP 41 (susceptible)	0.025	0.45	0.51	2.21	2.21	0.20	0.61/**0.51**	0.83/**0.68**	1.78/**1.73**	1.82/**1.79**
MRSP 67 (resistant)	0.9			0.24	0.29	11.2			**No efficacy** (0.0001/**0.0001**)	**No efficacy** (0.0003/**0.0002**)
MSSA B98 (susceptible)	0.056	0.45	0.51	2.12	2.14	0.4	0.61/**0.51**	0.83/**0.68**	1.64/**1.50**	1.79/**1.70**
MSSA B53 (resistant)	1.8			0.22	0.28	12.8			**No efficacy** (0.001/**0.0006**)	**No efficacy** (0.002/**0.001**)
*E. coli* 14L-1510 (susceptible)	0.0218	0.45	0.51	8.15	8.23	0.025	0.61/**0.51**	0.83/**0.68**	14.74/**14.29**	15.78/**14.98**
*E. coli* 10L-2253 (resistant)	2.8			0.081	0.11	14.4			**No efficacy** (0.002/**0.001**)	**No efficacy** (0.005/**0.0030**)

## Data Availability

The data presented in this study are available on request from the corresponding author.

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
