# Peer review of "Time-Kill Analysis of Canine Skin Pathogens: A Comparison of Pradofloxacin and Marbofloxacin"

_antibiotics, 2023, doi:10.3390/antibiotics12101548_

Round 1

Reviewer 1 Report

This manuscript is clearly written, from the introduction to the conclusion. I have read this manuscript several times, and have greatly appreciated its content. My comments on the form for a better understanding are mentioned in the attached file.

Author Response

The author team is very grateful for the time the reviewers spend reading the manuscript and making important suggestions.

Point by point review to each comments of both reviewers is provided below (black font : original comment, blue font: explanation from the authors, red font in “ “: changes in the text). Please also find attached a PDF with the reply to your useful comments. 

Kind regards

The author team

Reviewer 2 Report

The manuscript provides a detailed analysis of a study comparing two fluoroquinolone antibiotics, marbofloxacin and pradofloxacin, in veterinary dermatology. The study focuses on pharmacodynamic properties and their implications for clinical efficacy. Here are some suggestions to improve the manuscript quality for possible publications to the prestigious journal antibiotics:

1. The manuscript could benefit from clearer section headings and subheadings to help readers navigate the content more easily.

   2. please consider adding a brief introductory paragraph at the beginning of the Discussion section to provide an overview of the study's key findings and their significance.

3. The main objective of the study is not explicitly stated in the manuscript. It would be helpful to provide a concise statement of the research aim in the introduction.

4. While some abbreviations are explained, such as MIC and TKC, others like PK/PD are not defined. It's essential to include a list of abbreviations and their explanations for clarity.

5. In the "Materials and Methods" section, provide a clearer explanation of the procedures and methods used in the study, particularly in the subsections related to MIC measurement and pharmacodynamic data analysis.

6. Expand on the results of the covariate analysis and its implications for the study. How did the initial FQ susceptibility of isolates affect the pharmacodynamic parameters?

7. Provide more context and details on the in silico dose fractionation experiments, including the rationale for choosing specific dosing regimens and their relevance to clinical practice.

8. The manuscript would benefit from a more comprehensive interpretation of the results, particularly in the context of clinical implications. What do the findings suggest for the use of these antibiotics in veterinary practice?

9. Summarize the key findings and their practical implications in a concise conclusion section at the end of the manuscript.

10. Carefully proofread the manuscript for language and grammar errors to improve overall readability.

Author Response

The author team is very grateful for the time the reviewers spend reading the manuscript and making important suggestions.

Point by point review to each comments of both reviewers is provided below (black font : original comment, blue font: explanation from the authors, red font in “ “: changes in the text). Please also find attached the PDF with the reply to your useful comments. 

Kind regards

The author team

Reviewer 3 Report

I am sending my review comments to the manuscript Antibiotics 2647883 – entitled Time-kill analysis of canine skin pathogens. Comparison between pradofloxacin and marbofloxacin

Comments to the Authors

The manuscript prepared by Stefano Azzariti and co-workes report

1.      Conclusion is too short.

After careful reading of this paper I am of the opinion that the work is suitable to Antibiotics.

Author Response

The author team is very grateful for the time the reviewers spend reading the manuscript and making important suggestions.
Point by point review to each comments of both reviewers is provided below (black font : original comment, blue font: explanation from the authors, red font in “ “: changes in the text). Please also find attached a PDF with the reply to your useful comments.

Kind regards

The authors team

Round 2

Reviewer 2 Report

can be accepted

minor spelling and grammatical issues